# Effective Platinum-Copper Catalysts for Methanol Oxidation and Oxygen Reduction in Proton-Exchange Membrane Fuel Cell

**DOI:** 10.3390/nano10040742

**Published:** 2020-04-13

**Authors:** Vladislav Menshchikov, Anastasya Alekseenko, Vladimir Guterman, Andrey Nechitailov, Nadezhda Glebova, Aleksandr Tomasov, Olga Spiridonova, Sergey Belenov, Natalia Zelenina, Olga Safronenko

**Affiliations:** 1Chemistry Faculty, Southern Federal University, Rostov-on-Don 344090, Russia; men.vlad@mail.ru (V.M.); an-an-alekseenko@yandex.ru (A.A.); spiridonova_olga_alex@mail.ru (O.S.); serg1986chem@mail.ru (S.B.); ya.safronenko2014@yandex.ru (O.S.); 2Division of Solid State Electronics, Ioffe Institute, St. Petersburg 194021, Russia; aan.shuv@mail.ioffe.ru (A.N.); glebova@mail.ioffe.ru (N.G.); Alex.Tomasov@mail.ioffe.ru (A.T.); natochka56@mail.ru (N.Z.); 3PROMETHEUS R&D Ltd., Rostov-on-Don 344091, Russia

**Keywords:** platinum electrocatalyst, PtCu/C, oxygen electroreduction, methanol electrooxidation, catalyst activity, durability, fuel cell life tests, de-alloyed catalysts, PEM FC

## Abstract

The behavior of supported alloyed and de-alloyed platinum-copper catalysts, which contained 14–27% wt. of Pt, was studied in the reactions of methanol electrooxidation (MOR) and oxygen electroreduction (ORR) in 0.1 M HClO_4_ solutions. Alloyed PtCu*_x_*/C catalysts were prepared by a multistage sequential deposition of copper and platinum onto a Vulcan XC72 dispersed carbon support. De-alloyed PtCu*_x_*_−*y*_/C catalysts were prepared by PtCu*_x_*/C materials pretreatment in acid solutions. The effects of the catalysts initial composition and the acid treatment condition on their composition, structure, and catalytic activity in MOR and ORR were studied. Functional characteristics of platinum-copper catalysts were compared with those of commercial Pt/C catalysts when tested, both in an electrochemical cell and in H_2_/Air membrane-electrode assembly (MEA). It was shown that the acid pretreatment of platinum-copper catalysts practically does not have negative effect on their catalytic activity, but it reduces the amount of copper passing into the solution during the subsequent electrochemical study. The activity of platinum-copper catalysts in the MOR and the current-voltage characteristics of the H_2_/Air proton-exchange membrane fuel cell MEAs measured in the process of their life tests were much higher than those of the Pt/C catalysts.

## 1. Introduction

Low-temperature proton-exchange membrane fuel cells are promising sources of electrical energy that can be used in various types of devices, including automobiles, unmanned aerial vehicles, portable chargers, stationary units with a regular power supply, etc. [1,2,3,4]. The study and application of hydrogen-air (PEM FC) and methanol (DMFC) fuel cells are of particular interest [3,4,5,6,7]. During the operation of a hydrogen–air fuel cell with a proton exchange membrane, hydrogen is oxidized at the anode: H_2_ − 2e^−^ = 2H^+^, while two protons and two electrons are formed from one molecule. In a direct methanol fuel cell, an electrooxidation reaction of the simplest monohydric alcohol proceeds at the anode: CH_3_OH + H_2_O → CO_2_ + 6H^+^ + 6e^−^. Electrons pass through the external circuit, while protons migrate through a polymer electrolyte membrane (Nafion) [5,6,7]. Both electrons and protons reach the cathode, and next they react with the continuously supplied oxygen: O_2_ + 4H^+^ + 4e^−^ = 2H_2_O. High speed current-forming reactions require the presence of highly efficient electrocatalysts in the cathode and anode layers [8,9,10,11]. In an acidic environment, the best electrocatalyst for the above reactions is platinum, which is usually used in the form of nanoparticles deposited on the particles of dispersed carbon carriers [9,10,11,12].

It is well known that platinum alloying with the base metals such as Ni, Co, Cu, etc. contributes to an increase in the functional characteristics of catalysts in methanol electrooxidation (MOR) and oxygen electroreduction ORR [6,10,11,12,13,14]. The increase in the activity of bimetallic catalysts compared to Pt/C in the methanol electrooxidation reaction is usually explained by the bifunctional catalysis mechanism, in which OH groups adsorbed on the atoms of the doping component facilitate electrochemical desorption of the intermediate products of methanol oxidation from the Pt surface [6,15,16,17]. The optimization of catalysts for ORR is associated with an increase in the bond strength of Pt–O, which facilitates the adsorption of oxygen and intermediate oxygen-containing products (OH, HO_2_^−^, etc.) formed during its electroreduction. At the same time, an excessively strong adsorption interaction can hinder the desorption of reaction products [18,19]. Platinum doping with copper results in a decrease in the interatomic distance of Pt–Pt in the nanoparticles, as well as the change in the energy of free d-orbitals, which facilitates the adsorption of O_2_ on the surface of metal nanoparticles [11,18,19,20].

To reduce the content of the precious metal in the bimetallic catalyst, while increasing its activity in current-forming reactions, attempts have been made to optimize the composition and structure of platinum-containing nanoparticles [10,13,18,19,20,21,22,23,24]. Note that the features of the hierarchical structural organization of PtCu/C catalysts are determined not only by the shape and size of the metal nanoparticles, but also by their spatial distribution on the surface of the support, and have a crucial effect on the activity of catalysts in ORR and MOR [20,21,22,23]. The evaluation of some PtCu/C catalysts activity and stability in ORR and MOR obtained in electrochemical cells clearly indicates their high functional characteristics [20,21,22,24]. Unfortunately, the composition of bimetallic nanoparticles changes in the process of their functioning: there is a selective dissolution of atoms of the alloying component. When conducting studies in an electrochemical cell in the presence of a large amount of liquid electrolyte, the concentration of M*^z^*^+^ cations in the solution is low and it doesn’t have a noticeable effect on the electrode characteristics. A membrane–electrode assembly with a proton-conducting polymer electrolyte is another matter. Under the conditions of the membrane-electrode assembly (MEA) operation, the selective dissolution of the base metal from the nanoparticles is dangerous due to the possible poisoning of both the proton conducting polymer, which is a part of the catalytic layer, and the polymer membrane [24,25,26].

In our opinion, catalysts based on de-alloyed PtCu nanoparticles containing a relatively small amount of a “strongly bounded” copper could be very promising for use in membrane-electrode assemblies of the fuel cells. The direct synthesis of such catalysts is rather problematic, but one could try to obtain them from the copper rich PtCu*_x_*/C (*x* > 1) materials by treating them in acids. It is important to make it sure that, firstly, the predominant dissolution of Cu atoms allows the formation of a secondary Pt shell that protects the internal copper atoms from dissolution [22,27,28]. Secondly, it is necessary to determine, whether such de-alloyed PtCu*_x_*_−*y*_/C catalysts retain high stability and activity in ORR and MOR, which are characteristic of the previously studied alloyed PtCu*_x_*/C catalysts [27,28,29,30,31]. Note that uneven distribution of the doping component atoms in the “body” of the initial PtM nanoparticles obtained during the synthesis, as well as the changes in conditions of their de-alloying can apparently affect both the composition and activity/stability of the de-alloyed catalysts [21,22,32,33,34,35]. Thus, the search for the best PtCu compositions and structures of bimetallic nanoparticles, selection of the optimal conditions for the transformation of alloyed PtCu*_x_*/C catalysts into a de-alloyed state, as well as the study of the de-alloyed PtCu*_x_*_−*y*_/C catalysts behavior, proves to be timely and relevant.

This study is aimed at a comparative analysis of the functional characteristics of de-alloyed PtCu*_x_*_−*y*_/C catalysts obtained by the acid pretreatment of PtCu*_x_*/C materials, as well as their comparison with the characteristics of commercial Pt/C catalysts. It has been carried out not only in electrochemical cells with liquid electrolyte, but also in the MEAs of a hydrogen–air fuel cell with a proton exchange membrane.

## 2. Materials and Methods

Vulcan XC-72 carbon-supported platinum-copper catalysts were prepared by a four-stage synthesis in the liquid phase, sodium borohydride being used as a reducing agent as described in [22]. Initially, copper nanoparticles were deposited on a carbon support from a CuSO_4_ solution. At the second and third stages, the calculated amounts of copper (CuSO_4_) and platinum (H_2_PtCl_6_) precursors were added to the suspension. Subsequently, they were reduced with the excess sodium borohydride solution. At the fourth stage, formation of a platinum layer on the surface of nanoparticles was carried out by reducing Pt (IV) from the solution, which did not contain copper ions. The PtCu*_x_*/C suspension was filtered, and the catalyst was dried over P_2_O_5_. Heterogeneity of the components distribution (an increase in the platinum concentration from the center to the surface) in the platinum-copper nanoparticles, formed in accordance with the described synthesis technique, was proved in [22]. The composition of the obtained PtCu*_x_*/C materials samples, labeled S1–S5 below, corresponded to 1.7 ≤ *x* ≤ 2.9 values (see the “Results and Discussions” section). A portion of each of the synthesized catalysts was kept in solutions of different acids at 25 °C for 2–6 h with constant stirring. After being obtained in the liquid phase or treated in acid solutions, all materials were filtered, washed with the distilled water, dried at room temperature in a desiccator over P_2_O_5_. De-alloyed PtCu*_x_*_−*y*_/C materials, obtained from samples S1–S5 as a result of acid treatment, were labeled S1A–S5A, respectively. The more detailed information on the composition of specific materials and conditions of their acid treatment is given in the Results and Discussion section.

The ratio of platinum and copper in the platinum-copper catalysts was determined by the method of X-ray fluorescence analysis (XRFA) on a spectrometer with the total external reflection of X-ray radiation RFS-001 (Research Institute of Physics, SfedU, Rostov-on-Don, Russia). The mass fraction of metals was determined by the mass of the residue obtained after 40 min at 800 °C in air. For some samples the changes in the mass and the kinetics of high-temperature oxidation were studied by thermogravimetry. For this, the combined TGA/DSC/DTA NETZSCH STA 449 C analyzer was used. Oxidation was carried out in the atmosphere consisting of N_2_ (80%) and air (20%), in the temperature range from 20 to 750 °C at the heating rate of 10 °C/min and the gas flow rate of 20 mL/min, using corundum crucibles. When calculating the mass fractions of metals in PtCu/C samples, it was assumed that the non-combustible residue consists of Pt and CuO. The accuracy of mass fractions determination was ±0.4%.

The catalysts phase composition was determined by X-ray phase analysis (the voltage on the X-ray tube was 50 kV, the current was 150 μA, the time for the spectrum recording was 300 s, molybdenum anode was used). Diffractograms were taken on an ARL X‘TRA powder diffractometer with the Bragg-Brentano geometry (θ-θ), CuKα radiation (λ = 0.15405618 nm). Measurements were carried out at room temperature. Samples were thoroughly mixed and placed in a cuvette1.5 mm. in depth. The shooting was carried out in the angles range of 15–55 degrees with the step of 0.02 degrees and the speed of 8 to 0.5 degrees per minute, depending on the task set. The average crystallite size of the metal phase was determined using the Scherrer equation for a more intense reflection (111), as described in [22,36]. Electrochemical behavior of the catalysts in a standard three-electrode cell was studied by cyclic voltammetry (CV) at the solution temperature of 23 °C on an AFCBP bipotentiostat (Pine Research Instrumentation, Durham, CA, USA). A 0.1 M HClO_4_ solution (ChP, manufactured by Vecton, St. Petersburg, Russia) was used as an electrolyte. A saturated silver chloride electrode was used as a reference electrode, a platinum wire—as a counter electrode. All the potentials were given relative to the potential of the reversible hydrogen electrode (RHE). The studied electrode was a catalytic layer formed at the end of the glassy carbon disk electrode a drop of suspension being applied first. Before applying the suspension, the electrode surface was polished and then washed in isopropyl alcohol.

To obtain a catalyst suspension (catalytic “ink”), 900 μL of isopropyl alcohol and 100 μL of a 0.5% aqueous emulsion of Nafion^®^ polymer were added to 0.0060 g of each sample. Next, the suspension was dispersed by the ultrasound under cooling for 15 min. With continuous stirring, a 6 μL aliquot of “ink” was taken using a micropipette and applied to the end face of a polished and degreased glassy carbon electrode with an area of 0.196 cm^2^, the exact weight of the drop being recorded. After being dried, 7 μL of a 0.05% Nafion^®^ emulsion were applied to fix the catalytic layer, before the electrode was dried in air for another 15 min.

The studied electrode was standardized by means of 100 cycles of potential scanning in the range of values from 0.04 to 1.2 V at the rate of 200 mV/s. Then, 2 cyclic voltammograms (CV) were recorded on the fixed electrode at the potential sweep rate of 20 mV/s. The calculation of the electrochemically active surface area (ECSA) was performed for the second CV. For this purpose, the amount of electricity spent on electrochemical adsorption *Q_ad_* and desorption *Q_d_* of hydrogen was estimated as described in [22]. In some cases, ECSA was additionally measured by the oxidation of a chemisorbed CO monolayer, as described in [22]. In the latter case, the electrode was kept at a potential of 0.1 V in an electrolyte saturated with CO for 20 min. Then, the solution was purged with argon for 20 min, after that, two cyclic voltammograms were recorded, the purge not being stopped, and according to these voltammograms the calculation was performed. ECSA values are given with an accuracy of ±10%.

While studying the catalysts activity in MOR, methanol and perchloric acid were added to the electrochemical cell, thereby obtaining a solution of 0.1 M HClO_4_ + 0.5 M CH_3_OH. Cyclic voltammograms were recorded in the potential range of 0.04–1.3 V with a potential sweep rate of 20 mV/s. Chronoamperograms were measured at the potential of 0.70 V. The measurements were carried out in the argon atmosphere.

To evaluate the catalysts activity in ORR, a 0.1 M HClO_4_ solution was saturated with oxygen for 1 h, after that a series of voltammograms was measured in the range from 0.12 to 1.19 V with a linear potential sweep at the speed of 20 mV/s at the electrode rotation speeds of 400, 900, 1600, and 2500 rpm. To take into account the contribution of the ohmic potential drop and non-ORR related processes, the voltammograms, obtained at the potential scanned towards more positive values were normalized according to the generally accepted methods [37,38,39,40]. For this, the potential of the electrode under study was refined by the formula: *E = E*_set_
*– It* × *R* where: *E*_set_ is the set value of the potential, *I_t_* × *R* is the ohmic potential drop equal to the product of the current strength by the resistance *(R)* of the solution layer between the reference electrode and the studied electrode, which in this case was 23 ohms. This resistance value is in good agreement with the published data [38]. The contribution of the processes, occurring on the electrode in the oxygen-free solution (Ar atmosphere), was taken into account by subtracting from the voltammogram a similar curve recorded on the same electrode during measurements in the Ar atmosphere: *(I(O_2_) − I(Ar))*, as described in [39,40]. The catalytic activity of the catalysts in the ORR (kinetic current) was determined by the normalized voltammograms, taking into account the contribution of mass transfer under the conditions of rotating disk electrode (RDE) [39,40]. The kinetic current was calculated according to the Koutetsky–Levich equation: 1*/j =* 1*/j_k_ +* 1*/j_d_*, where *j* is the experimentally measured current, *j_d_* is the diffusion current, and *j_k_* is the kinetic current. Kinetic currents were calculated for the potential of 0.90 V (RHE).

It is well known that the ECSA values and the values of the catalyst activity depend on the composition, structure, and method of the catalytic layer deposition [37,40,41,42]. Therefore, the electrodes, on which similar catalytic layers were formed by the same method but with commercial Pt/C HiSPEC3000 and HiSPEC4000 (Johnson Matthey, Swindon, UK) electrocatalysts containing 20% and 40% wt. platinum, respectively, were used as the reference samples.

The tests of electrocatalysts in the membrane-electrode assembly were carried out as follows. To form the cathode and anode layers, the studied catalyst was applied in the necessary amount to achieve platinum loading of 0.4 mg/cm^2^. A commercial Nafion solution (DE-2020, 20% by weight) was used.

Technological operations to prepare the dispersion of the electrode material, required for the subsequent fabrication of the MEA, included two stages: mechanical and ultrasonic dispersion of a mixture of precisely weighed components in an isopropanol-water mixture. The volume ratio of the liquid components of isopropanol to water was ~1: 1. The ratio of the masses in the solid and liquid phases in the final dispersion was in the range 1:40–1:80.

Mechanical dispersion was performed on a magnetic mixer of the Milaform MM-5M type for ~0.5 h with an arm rotation speed of ~400 rpm until a visually homogeneous (without visible lumps) mass was obtained. Subsequent ultrasonic dispersion was carried out in a Branson 3510 ultrasonic bath for 40–100 h to obtain a homogeneous dispersion that did not delaminate for one minute.

Membrane–electrode assemblies were made by applying a uniform dispersion of components directly to the proton-conducting membrane through a stainless-steel mask. Before applying the electrode material, the membrane was kept in 0.5 M sulfuric acid for 15 min at temperature 70–80 °C, followed by five times washing with water. The electrodes were made by smearing the dispersion of components in an isopropanol–water mixture at room temperature onto a 50 μm (46–50 μm) thick Nafion proton-conducting membrane pre-treated at 85 °C. The amount of supported catalyst was controlled gravimetrically. Before conducting electrochemical measurements, the MEA was kept in 0.5 M sulfuric acid for 15 min at temperature 70–80 °C, followed by five times washing with water. After that, the MEA was placed in a standard measuring cell (FC-05-02, ElectroChem, Inc., Woburn, MA, USA) with graphite collector electrodes. Toray060 standard carbon paper was used as the gas diffusion layer.

The change in the MEA characteristics during aging was monitored by the current-voltage characteristics. We used the two-electrode method, potentiostat P-150 (LLC Electrochemical Instruments, Moscow, Russia). Potential sweep rate −10 mV/s. Before starting the main measurements, the MEA was activated as described in [43]. MEA aging was carried out at room temperature and atmospheric pressure for a given number of cycles (0, 100, 300, 1000, etc.) of a voltage scan in the range 0.6–1.0 V with a potential scan rate of 50 mV/s. Wet (≈100%) N_2_ and H_2_ were applied to the electrodes. An aging electrode was supplied with N_2_. The MEA aging regime corresponded to that of the DOE protocols (Table A-1 in [44]).

## 3. Results and Discussion

### 3.1. Methanol Electro-Oxidation at the Alloyed and De-Alloyed Platinum-Copper Catalysts

The initial alloys and the platinum-copper catalysts obtained from them after two-hour treatment in 1 M HNO_3_ at 25 °C (de-alloyed), which contained from 18.5 to 26.6% wt. of platinum, were studied as catalysts for the methanol electro-oxidation (Table 1).

The X-ray diffraction patterns of platinum-copper materials contain reflections corresponding to the phases of carbon and platinum (Figure 1). In this case, platinum characteristic reflections are shifted toward large values of the 2 theta angles, this being a consequence of a Pt–Cu solid solution formation. A significant difference in the composition of the initial S1–S3 samples had practically no effect on the position of the characteristic reflections of the X-ray diffraction pattern, in particular, on 2 theta angles of the characteristic maximum 111 (Table 1, Figure 1b). Apparently, a larger or a smaller fraction of copper in these samples is contained in X-ray amorphous oxides [45], while the composition of the metal component of various catalysts is indeed similar. At the same time, the 2-theta angle of the maximum 111 in the diffraction patterns decreases with the transition from the starting materials (S1, S2, S3) to the de-alloyed catalysts obtained from them (Table 1, Figure 1). This indicates a decrease of the copper content in the catalysts metal component and correlates with the obtained results, when their composition was determined (Table 1).

The average crystallite diameters in the studied bimetallic materials, calculated by the Scherrer formula, are from 2.7 to 3.0 nm (Table 1). Given the accuracy of the measurements, this indicates the absence of a significant effect of the initial composition and acid treatment on their size. At the same time, it should be noted that the application of the Scherrer equation to calculate the average size of bimetallic nanoparticles may provide results which are not quite accurate, since such calculation does not take into account the contribution of the inhomogeneous structure of nanoparticles to the broadening of reflection 111 [45].

Acid treatment of all the initial PtCu*_x_*/C samples leads to the dissolution of a significant amount of copper (Table 1). A decrease in the concentration of copper in the samples cannot be associated only with the dissolution of its amorphized oxides, since judging by the data of the X-ray diffractometry, the content of metallic copper in the solid solution based on platinum, decreases. This leads to the observed decrease in the 2-theta angle for the maximum (111) during the transition from the initial to the acid-treated samples (Table 1, Figure 1). Note that despite the significant difference in the initial compositions of the platinum-copper catalysts, the compositions of the three de-alloyed samples are similar to each other: Pt_53_Cu_47_–Pt_56_Cu_44_. Thus, the amount of “firmly bound” copper, whose selective dissolution from platinum–copper nanoparticles does not occur at the stage of acid treatment, slightly depends on the initial composition of the materials, obtained by the synthesis method described above.

The behavior of the initial and de-alloyed catalysts in the process of electrochemical standardization has significant differences. CVs of S1–S3 samples during the first few cycles show anodic maxima in the potential range 0.25–0.35 V, associated with the dissolution of copper from the intrinsic phase (Figure 2) [17,24]. For the samples of de-alloyed catalysts S1A–S3A, similar maxima on CVs were not observed. It should be noted that the materials surface development in the “as prepared” state continues during a greater number of cycles than for catalysts treated in acid (Figure 2). This indirectly indicates a long-term reorganization of the nanoparticles surface, due to a more intensive anodic dissolution of copper atoms. According to the published data [17,21], the anodic dissolution of copper from the Pt–Cu solid solution mainly occurs in the potential range 0.60–0.85 V and is reflected by a gradually decreasing local maximum on the CVs at the above potentials (Figure 2, samples S1–S3).

It was found that during the standardization stage the copper content in all studied catalysts (alloyed and de-alloyed) decreased due to its selective dissolution, the composition of the materials, as a result, being approximately the same Pt_63_Cu_37_–Pt_67_Cu_33_ (Table 1). Moreover, 70% to 83% and 33 to 45% of the copper content are dissolved from the alloyed and de-alloyed catalysts, respectively. Given the lower content of the alloying component in de-alloyed catalysts (Table 1), we can assume a significant decrease in the concentration of copper (II) cations in the electrolyte and, therefore, a decrease in their possible negative effect on Nafion. Nevertheless, in order to exclude the influence of copper (II) cations, which were present in the solution, on the behavior of the catalysts, prior to further electrochemical studies, the electrolyte was replaced with a freshly prepared 0.1 M HClO_4_ solution, which was then saturated with argon for 20 min.

Figure 3a,c shows cyclic voltammograms of the standardized catalysts. They have the form characteristic of Pt/C and PtM/C electrocatalysts [17,21,24]. The ECSA values of PtCu/C materials, determined by the areas of electrochemical adsorption/desorption of atomic hydrogen and the oxidation peaks of the monolayer of chemisorbed CO, are in the range from 30 to 40 m^2^/g (Pt) (Table 2), which is much lower than that of the commercial Pt/C sample. The lower ECSA values of the platinum–copper catalysts, compared to the commercial Pt/C sample, are largely due to the larger average crystallite size (Table 1), as well as the agglomeration (aggregation) of the nanoparticles.

The acid treatment of S1–S3 catalysts does not lead to a considerable change in the platinum ECSA (Table 2). Apparently, the stage of electrochemical standardization, leading to the formation of catalysts with approximately the same composition, eliminates the differences in ECSA values that could occur in the initial and de-alloyed samples.

The oxidation of chemisorbed CO on all platinum-copper catalysts begins at the potentials lower than those on Pt/C, and ends in a narrower range of potentials, approximately 0.63–0.88 V (Figure 3b,d). Bimodal maxima of CO oxidation (Figure 3b,d) can be related to heterogeneity of the composition and structure of bimetallic nanoparticles, as well as their size dispersion [30].

Electrocatalysts activity in the MOR was determined by the methods of cyclic voltammetry and chronopotentiometry. Using the CVs measured in 0.5 M solutions of CH_3_OH, we calculated the specific amount of electricity *Q*_CH3OH_ (Cl/g(Pt)) spent on methanol oxidation in the direct course of potential sweep (Figure 4a,c, Table 2). Despite the smaller ECSA values for all platinum-copper catalysts, *Q*_CH3OH_ is greater than that for Pt/C (Table 2).

The same is true for the maximum currents of methanol oxidation on CVs: for bimetallic catalysts they are essentially higher than for Pt/C (Figure 4 and Table 2). The potentials for the onset of methanol oxidation, determined for platinum-copper catalysts by the corresponding CVs of the forward scan, are approximately 150–200 mV less than for commercial Pt/C. Despite a considerably higher methanol oxidation rate and, possibly, a higher concentration of intermediate reaction products on the electrode surface, platinum-copper catalysts, as a whole, showed higher current values when chronoamperograms were recorded at the potential of 0.7 V (Table 2, Figure 4b,d) compared to Pt/C. Together with a lower oxidation potential of a monolayer of chemisorbed CO (Figure 3b,d), this indicates a higher tolerance of bimetallic catalysts to intermediate products of methanol oxidation. The least activity in MOR was demonstrated by S3 and S3A catalysts.

It should be noted that de-alloyed PtCu*_x_*_−*y*_/C catalysts are not inferior in activity to MOR electrocatalysts, which were deposited on a glass-graphite electrode in the “as-prepared” state and initially contained a significantly larger amount of copper. Obviously, the reason for their similar behavior is the proximity of the composition/structure of the platinum-copper catalysts, formed after a portion of copper being selectively dissolved due to a chemical and/or electrochemical treatment of the initial materials (Table 1) [45]. A comparative analysis of the electrochemical behavior of the platinum-copper catalysts in MOR (Figure 4, Table 2) suggests a noticeably higher activity of samples S1, S1A, S2 and S2A compared to S3 and S3A: initial S1 and S2, and de-alloyed S1A, S2A catalysts exhibit higher values of *Q*_CH3OH_ (Table 2), electric current of “methanol maximum” on CVs (Figure 4a,c) and currents on chronoamperograms of methanol oxidation (Figure 4b,d). This difference in the catalysts behavior can be partially due to the difference in the composition of the metal component after electrochemical standardization: for example, S3A sample contains slightly less copper than S1A and S2A (see Table 1). In this case, as was noted above*,* reliable differences in the ECSA values of different platinum-copper catalysts were not established (Table 2). In our opinion, one should take into account possible differences in the character of atoms localization of a “firmly bound” copper in the standardized nanoparticles, their surface being enriched in platinum. The smaller the thickness of the secondary platinum shell, the stronger the promoting effect of the alloying component atoms on the activity of platinum in electrochemical reactions can be expressed.

This study has shown that platinum-copper catalysts exhibit much higher mass activity (A/g(Pt)) and specific activity (A/m^2^) in MOR compared to the commercial Pt/C material, despite a significantly lower ECSA. The pretreatment of catalysts in the nitric acid promotes leaching of a larger copper fraction from the nanoparticles, but does not lead to a decrease in the activity of catalysts in MOR. Therefore, such a procedure can be used for the pretreatment of catalysts to reduce the subsequent pollution of MEAs with copper cations.

### 3.2. Electroreduction of Oxygen on Alloyed and De-Alloyed Platinum-Copper Catalysts

The next step in our research was to test the hypothesis that changes in the composition and structure of the platinum-copper nanoparticles, and, as a result, their activity in the electroreduction of oxygen may depend on the processing conditions in acids, in particular, on the nature and concentration of the acid. At this stage, the Pt_31_Cu_69_/C (S4) catalyst with a low platinum content (about 14 wt%) was used as the initial sample. Equal portions of this material were subjected to a 6-h treatment in solutions of different acids at room temperature ~23 °C (Table 3). Then the composition, structural characteristics (Table 3) and the electrochemical behavior of the obtained de-alloyed catalysts were studied.

The initial S4 sample treatment in acids leads to a selective dissolution of a greater amount of copper and, apparently, to a small amount of platinum. As a result, the composition of the metal component varies from Pt_31_Cu_69_ (S4) to Pt_57_Cu_43_–Pt_58_Cu_42_ for most of the de-alloyed samples (Table 3). The total mass fraction of metals also changes accordingly: from 23.4% mass (S4) up to 16.6–17.8% mass for pre-treated samples (Table 3). Note that the sample treated in a 5 M solution of the nitric acid is characterized by the lowest residual copper content (Pt_63_Cu_37_) in comparison with other de-alloyed materials (Table 3). This is due to the high concentration and aggressiveness of the acid, used for pretreatment. It is also impossible to exclude completely the possibility of corrosion of the carbon carrier, which occurs under the conditions of acid treatment, especially when S4-5N is obtained.

The thermograms of high-temperature oxidation of PtCu/C materials (Figure 5) have the form characteristic of platinum-carbon catalysts [46,47]. In this case, the most rapid oxidation of the carbon carrier is observed for the initial Pt_31_Cu_69_/C (A4) sample and the Pt_57_Cu_43_/C (A4-1N) material subjected to treatment in 1 M nitric acid. Taking into account the data on the composition and size of the nanoparticles given in Table 3, as well as the results of Ref. [46], one can make an assumption that a longer or shorter duration (temperature range) of intense combustion of the studied materials may be due to a more (rapid oxidation) or less (slow oxidation) ordered distribution of the nanoparticles over the surface of the carbon carrier microparticles.

A change in the composition of the metal component of the catalysts obtained after acid treatment is also confirmed by the results of X-ray diffractometry (Figure 6a,b): the reflection maximum 111 for all post-treated samples (Figure 6b) shifts to lower 2 theta angles compared to the maximum for the starting material S4, which is due to a decrease of copper concentration in the nanoparticles.

All the catalysts obtained after the acid treatment are characterized by a decrease in the average crystallite sizes from the initial 2.8 to 2.6–1.8 nm (Table 3). The most significant decrease in this parameter compared to the initial sample is also observed for sample S4-5N obtained after processing the starting material in the most aggressive medium. Probably under conditions of rapid structural restructuring caused by the intense corrosion of copper, the crystallites of this material acquire the highest defectiveness, which makes an additional contribution to the peak broadening in the X-ray diffraction pattern (Figure 6b) and underestimates their average size, calculated by the Scherrer equation.

CVs of the initial (S4) and acid-treated electrocatalysts recorded during standardization have the form characteristic of platinum nanoparticles or its alloys deposited on a carbon support (Figure 7) [17,21,24]. In the hydrogen region of CV (0.03–0.3 V), a gradual increase in currents is observed due to the cleaning and development of the surface of the PtCu/C materials, as was noted above (Figure 7). Moreover, there are no peaks in the potential range 0.25–0.35 V associated with the dissolution of copper from the intrinsic phase [17,24] on the anode CV branches of the initial (S4) and all de-alloyed PtCu/C materials (Figure 7). This indirectly indicates the absence of direct contact of the copper phase with the electrolyte. Note that the development of the surface of sample S4 (“as prepared” state) is much more pronounced (continues for a larger number of cycles) than for PtCu/C catalysts treated in acid (Figure 7). Obviously, this is largely due to a more intense anodic dissolution of copper atoms. The differences in the character of changes in the anode part of CV observed for S4, as compared to de-alloyed catalysts at potentials of about 1 V (Figure 7), are also apparently due to a significant difference in the surface composition of nanoparticles.

Determination of the catalysts ECSA was carried out by the amount of electricity spent on the electrochemical adsorption and desorption of atomic hydrogen during CV registration. The highest ECSA value among platinum-copper catalysts was observed for the sample S4-1N—41.2 m^2^/g (Pt), the lowest, 35.7 and 35.2 m^2^/g (Pt) was for S4-1Cl and S4-1S, respectively (Table 4). Unfortunately, taking into account the accuracy of the ECSA values determination (Table 4), the measurements result in itself does not allow us to speak of a reliable dependence of its values on the conditions for processing the S4 catalyst in acids. At the same time, the ECSA of the commercial Pt/C reference sample JM20 is approximately 2 times higher than that for platinum-copper samples of the S4 series (Table 4), being similar to what was previously established for the samples of the S1–S3 series (Table 2).

From the comparison of the metal component composition of the platinum-copper catalysts before (Table 3) and after measurement completion (Table 4), it follows that in the process of conducting electrochemical measurements (standardization of electrodes, ECSA determination, evaluation of the catalysts activity in ORR at different speeds of the disk electrode rotation) all the studied samples lose some amount of copper. The loss is from 77% (catalyst S4) to 28–39% (acid pretreated materials) of the initial copper concentration in the catalysts deposited on the electrode. In this case, the lowest degree of copper recovery is characteristic of catalysts S4-1N and S4-5N, treated in the nitric acid solutions (Table 3 and Table 4).

Electrocatalysts activity in the reaction of oxygen electroreduction (ORR) was studied by recording voltammograms with a linear scan of potential for RDE in an oxygen atmosphere. Normalized, as described in the Experimental section, linear sweep voltammograms (LSVs) re shown in Figure 8a. The relationship between the current strength and rotation speed was analyzed at potential 0.90 V in the Koutetsky-Levich coordinates (Figure 8b).

The voltammograms obtained for all studied PtCu/C catalysts (Figure 8a) are characterized by close values of a half-wave potential of the oxygen reduction reaction (Table 4). The analysis of the straight-line dependences in the coordinates 1/*I—*ω^−0.5^ (Figure 8b) made it possible to calculate the kinetic currents of ORR and confirmed the four-electron reaction mechanism characteristic of platinum (the number of electrons n ~4) (Table 4). The specific activity of platinum-copper catalysts in ORR was approximately 1.7–2.6 times higher than that of JM20 (Table 4). The order in the increase of catalysts mass–activity can be presented as follows: S4-1S < S4-1Cl ≤ S4-5N < JM20 ≈ S4 < S4-1N (Table 4). The combination of the minimum specific and relatively high mass activity observed for JM20 is due to a higher ECSA of this catalyst compared to platinum-copper materials. Note that the highest residual concentration of the copper atoms, mainly localized in the inner layers of PtCu nanoparticles, may be one of the reasons for the higher activity of S4-1N and S4 samples in ORR.

It is important to note that the selective dissolution of copper during standardization of the electrodes and electrochemical measurements that followed, takes place on all PtCu/C catalysts. However, for the samples previously treated in acid, as was previously established for samples of the S1–S3 series, the amount of copper passing into the solution, and the change in the composition for all de-alloyed catalysts obtained from S4, is much less pronounced than for the initialPt_31_Cu_69_/C (S4) catalyst (compare the composition of the catalysts before (Table 3) and after (Table 4) electrochemical measurements). This confirms our assumption that an “acid treatment” can be used to wash out “loosely bound” copper from the platinum catalysts, as its presence makes their use in the MEA problematic.

It is known that the catalysts exhibiting high ORR activity in the electrochemical cell do not always demonstrate similar results when tested in the MEA [1,2,3]. In the case of platinum-copper catalysts, the negative effect can also be associated with the poisoning of Nafion by copper cations, formed as a result of its selective dissolution from the bimetallic nanoparticles [24,25,26]. Based on the results of electrochemical behavior of alloyed/de-alloy PtCu/C materials, we synthesized the platinum–copper catalyst S5, an analogue of S4, which contained a greater loading of metals. Through a three-hour treatment of portions of this material in 1 M sulfuric and nitric acids at 25 °C, de-alloyed samples of similar composition—S5N and S5S, respectively, were obtained. The mass fraction of platinum in these catalysts was 18.3 ± 0.5% wt., and the ratio of platinum and copper corresponded to the formula Pt_53_Cu_47_. These catalysts were used to form the cathodic and anodic catalytic layers during the assembly of H_2_/air PEMFC MEAs. The MEAs were assembled using a similar technology based on the commercial Pt/C PM40 catalyst (40% wt. Pt, PROMETHEUS R&D, Rostov-on-Don, Russia), which previously demonstrated high functional characteristics [48]. The conditions for conducting MEAs life tests, which included 5000 cycles of potential changes, are described in the Experimental section, and some test results are presented in Figure 9 and Figure 10.

Initially, all three MEAs showed similar current-voltage and watt-ampere characteristics (Figure 9a). However, as the cycling progressed, a decrease in the characteristics of the Pt/C catalyst based MEAs was observed, while both MEAs with PtCu/C catalysts during 2000 cycles even slightly increased their characteristics (Figure 9b,c). After 2800 sweep cycles, the Pt/C-based MEA tests had to be stopped due to electrode degradation. At the same time, for the MEA based on A5S after 5000 cycles of potential scanning, only a small decrease in power was recorded, while for the MEA containing A5N, the current-voltage characteristics even slightly increased (Figure 10). Thus, the test results of electrocatalysts in the MEAs (Figure 9 and Figure 10) correlate well with the results obtained by studying their activity in ORR (Table 2 and Table 4) and stability, when tested in an electrochemical cell [49]. The negative influence of Cu^2+^ cations formed during the operation of the de-alloyed PtCu/C electrocatalysts on proton conductivity (membrane resistance) of Nafion was not recorded.

## 4. Conclusions

PtCu*_x_*/C catalysts of various compositions (1.7 ≤ *x* ≤ 2.9), containing bimetallic nanoparticles in which the platinum concentration increases from the center to the surface of the particles were synthesized by sequential chemical reduction of copper, and next, of copper and platinum from the solutions of their precursors. By treating portions of PtCu*_x_*/C materials in the solutions of acids of different nature and concentration, the de-alloyed PtCu*_x_*_−*y*_/C catalysts (13.8–26.4% wt. Pt) with a reduced copper content (x − y = 0.58–0.90) were obtained.

Both alloyed and de-alloyed catalysts are characterized by a decrease in the copper content in their composition as a result of electrochemical standardization of their surface. Nevertheless, for de-alloyed materials, the amount of copper transferred to the electrolyte is considerably less. The composition of the post-treated PtCu*_x_*_−*y*_/C catalysts weakly depends on their initial composition and, to a greater extent, on the acid treatment conditions: the higher the acid concentration and the processing time, the lower the residual copper content is. We have not established a significant effect of the initial composition and pretreatment conditions of the catalysts on their ECSA, measured after standardization. The ECSA values of the alloyed and de-alloyed electrocatalysts measured after standardization of the electrodes ranged from 31 to 43 m^2^/g (Pt), while the ECSA values of the commercial Pt/C catalyst (40% wt. Pt) used for comparison, were about 80 m^2^/g (Pt).

Despite the lower values of ECSA, the supported platinum-copper catalysts exhibit significantly higher mass activity and specific activity in MOR compared to the commercial Pt/C electrocatalyst. The difference in mass activity of the platinum-copper and platinum catalysts in ORR is not that crucial, however, some samples of PtCu/C catalysts were also more active than Pt/C.

Pretreatment of PtCu/C catalysts in acids leads to the leaching of a significant fraction of copper from the nanoparticles, but it does not cause a decrease in their activity in MOR and ORR. The fact is that the electrochemical standardization of the catalysts causes an even higher degree of extraction of a “weakly bound” copper from the nanoparticles, therefore their initial composition does not play the leading role. At the same time, the conditions of acid pretreatment of the initial PtCu*_x_*/C catalysts can have some effect on the change in the composition/structure of the nanoparticles and, as a consequence, on their electrochemical behavior. For example, the highest activity in ORR was demonstrated by a de-alloyed catalyst pretreated in 1 M HNO_3_. The influence of acid treatment conditions on the structural and electrochemical characteristics of platinum-copper catalysts was also noted by Hodnik et al. [28,50]. Therefore, acid treatment conditions can be optimized.

The combination of high activity in ORR and satisfactory resistance to the selective dissolution of copper made it possible to conduct life tests of de-alloyed PtCu/C electrocatalysts in the H_2_/Air MEA. At the initial stage of stress testing, the current-voltage characteristics of MEAs containing bimetallic catalysts and MEA, based on commercial Pt/C, were close but during stress testing the platinum-copper catalysts showed much higher stability. As a result, the life tests of the MEA with a Pt/C catalyst were completed after 2800 cycles of potential change; when de-alloyed PtCu/C catalysts were used, the MEAs successfully withstood 5000 cycles, and one of them even slightly increased its power. These results correlate well with the data on the increased stability of PtCu/C catalysts, which were obtained in [49] from stress testing of PtCu/C and Pt/C catalysts in an electrochemical cell.

Note that we did not detect the negative effect of copper cations formed during the stress testing of the de-alloyed catalysts on the current-voltage characteristics of the MEA. This fact confirms the assumption that the acid pretreatment of PtCu_x_/C materials allows one to obtain the de-alloyed PtCu*_x_*_−*y*_/C catalysts that possess not only high durability, but also stability, which is sufficient from the point of the copper selective dissolution.

The results of the study are in good agreement with the data, that we have previously obtained [22,29,49] and with those published in literature [11,27,28]. At the same time, they demonstrate new opportunities to control the composition, structure, and functional characteristics of the de-alloyed PtCu*_x_*_−*y*_/C catalysts. Pretreatment of the platinum-copper catalysts in acids does not practically reduce their activity in ORR and MOR, but when the “excess” of copper is removed, resistance to its selective dissolution significantly increases. As a result, the de-alloyed PtCu*_x_*_−*y*_/C catalysts exhibit high activity and stability during the life testing of MEAs and do not cause proton-conducting polymer poisoning. In our opinion, the results obtained also confirm the feasibility of further de-alloyed PtCu*_x_*_−*y*_/C catalysts testing in the fuel cell MEAs.

## Figures and Tables

**Figure 1 nanomaterials-10-00742-f001:**
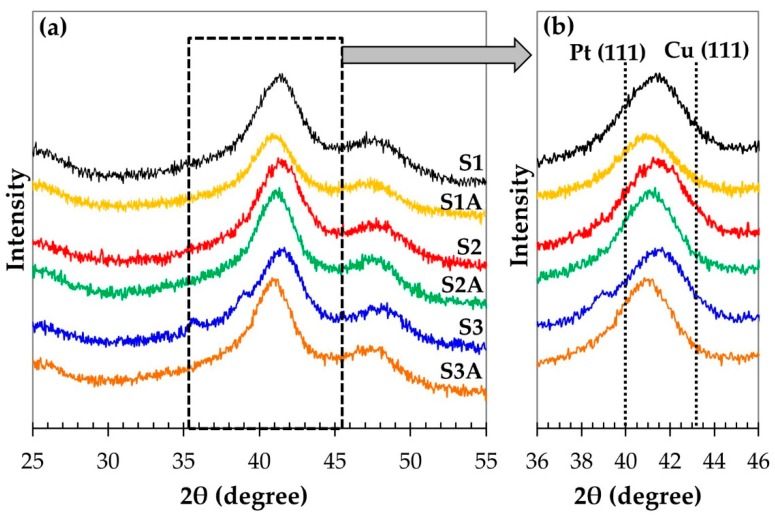
(**a**) X-ray diffraction patterns of the “as-prepared” S1, S2, S3 PtCu*_x_*/C and de-alloyed S1A, S2A, S3A PtCu*_x_*_−*y*_/C catalysts. (**b**) In the inset (on the right)—an enlarged fragment of the diffractogram area, highlighted by the dotted line in Figure (**a**).

**Figure 2 nanomaterials-10-00742-f002:**
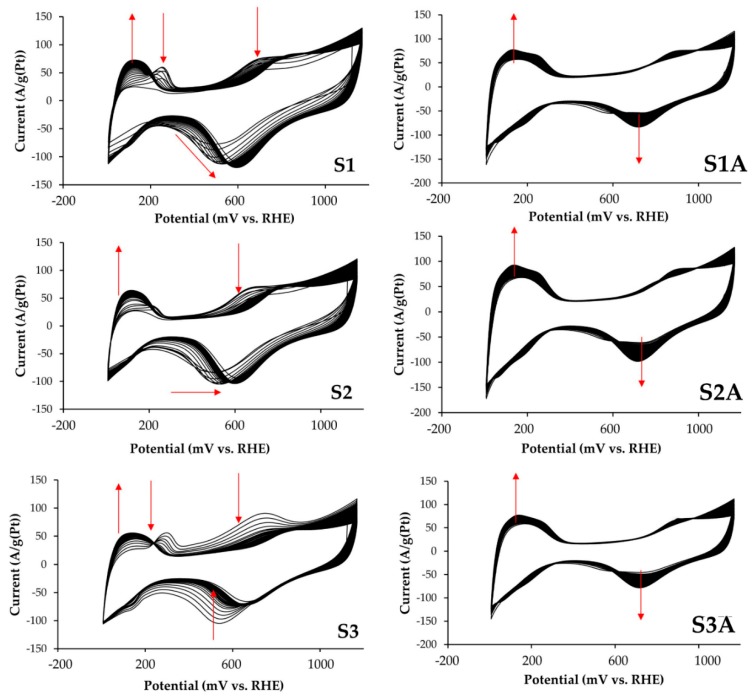
Cyclic voltammograms of catalysts in the standardization process (100 cycles). The electrolyte solution is 0.1 M HClO_4_ saturated with argon at atmospheric pressure. The sweep rate of potential is 200 mV/s.

**Figure 3 nanomaterials-10-00742-f003:**
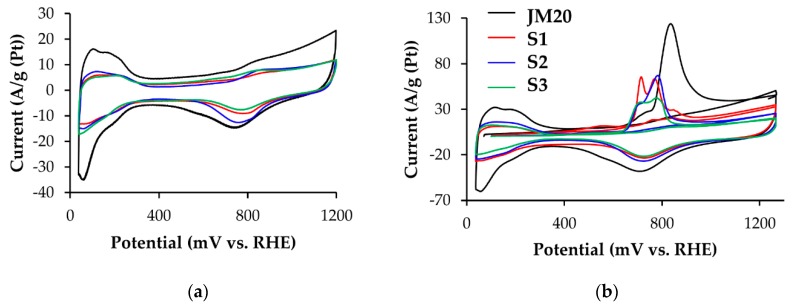
Cyclic voltammograms of electrocatalysts in the argon atmosphere (**a**,**c**) after completion of standardization and after CO purging (**b**,**d**). Commercial Pt/C (JM20) and initial PtCu*_x_*/C materials—(**a**,**b**); commercial Pt/C (JM20) and de-alloyed PtCu*_x_*_−*y*_/C materials.

**Figure 4 nanomaterials-10-00742-f004:**
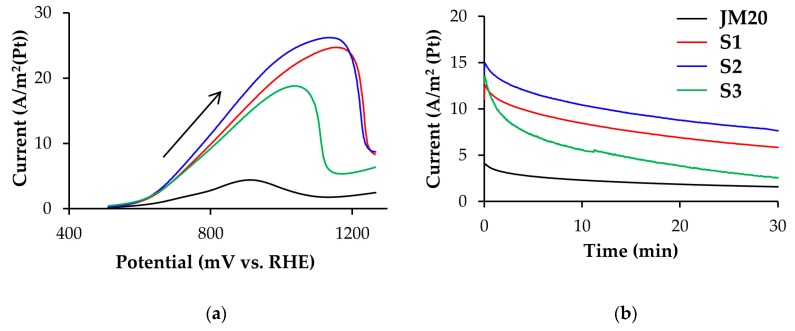
(**a**,**c**) Cyclic voltammograms and (**b**,**d**) chronoamperograms of commercial Pt/C (JM20) and the studied (**a**,**b**) alloyed and (**c**,**d**) de-alloyed PtCu/C catalysts. The currents are normalized to ECSA. The potential sweep rate during CVs registration is 20 mV/s. The electrolyte is a solution of 0.1 M HClO_4_ + 0.5 M CH_3_OH saturated with Ar at atmospheric pressure.

**Figure 5 nanomaterials-10-00742-f005:**
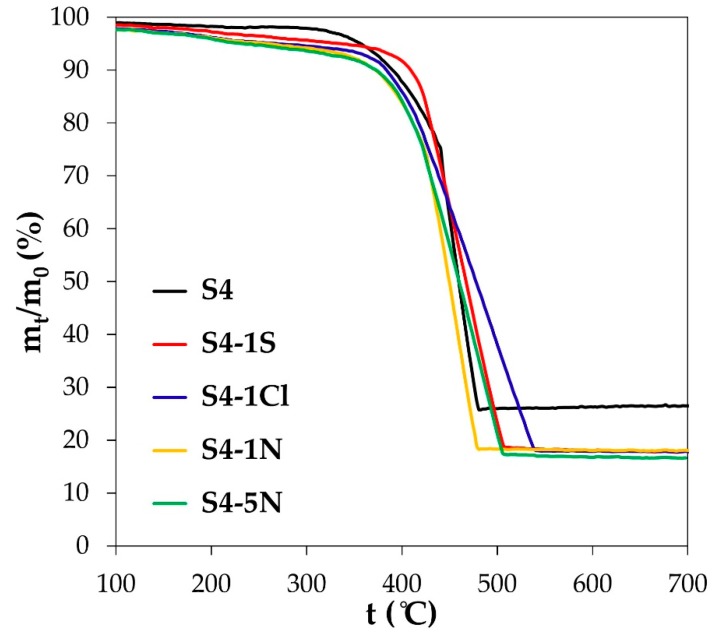
Thermograms of high-temperature oxidation of metal-carbon materials (marking corresponds to Table 2).

**Figure 6 nanomaterials-10-00742-f006:**
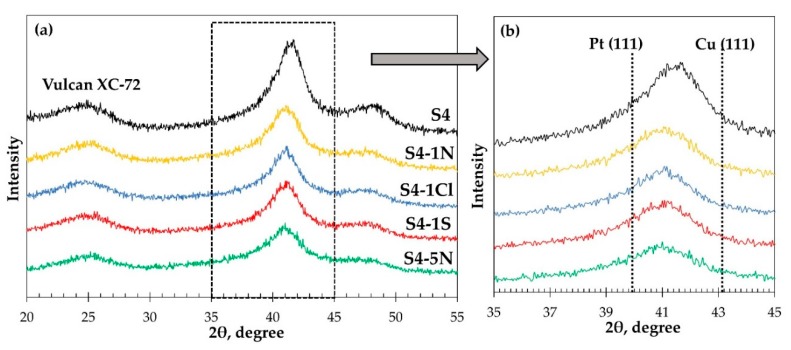
(**a**)X-ray diffraction patterns of the initial metal-carbon material and the material obtained after its treatment in acids. (**b**) In the inset (on the right)—an enlarged fragment of the diffractogram area, highlighted by the dotted line in Figure (**a**).

**Figure 7 nanomaterials-10-00742-f007:**
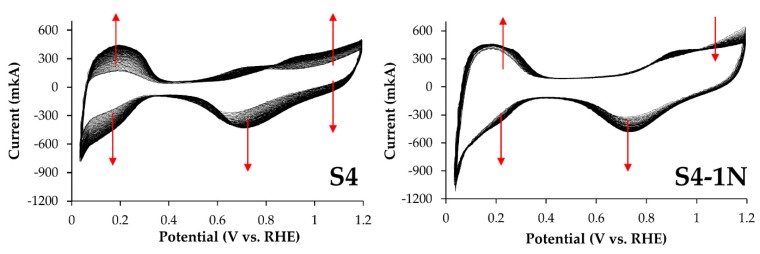
Cyclic voltammograms of PtCu/C electrocatalysts corresponding to 1, 25, 50, 75 and 100 d standardization cycles. The arrows indicate the direction of CV displacement in the standardization process.

**Figure 8 nanomaterials-10-00742-f008:**
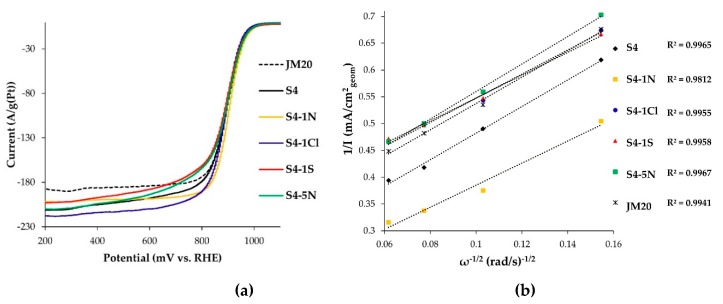
(**a**) LSV curves of electroreduction of oxygen on different catalysts. Disk rotation speed of 1600 rpm. (**b**) Dependencies 1/*I* on ω^−1/2^ at a potential of 0.90 V. Samples: 1—S4, 2—S4-1N, 3—S4-1Cl, 4—S4-1S, 5—S4-5N, 6—JM20.

**Figure 9 nanomaterials-10-00742-f009:**
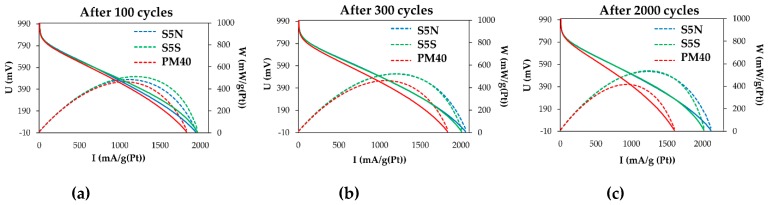
Polarization and power density curves of MEAs after 100 (**a**), 300 (**b**) and 2000 (**c**) stress testing cycles. Catalyst layers were prepared using PM40, S5N, and S5S catalysts. The loading of platinum in each catalytic layer was of 0.4 mg/cm^2^.

**Figure 10 nanomaterials-10-00742-f010:**
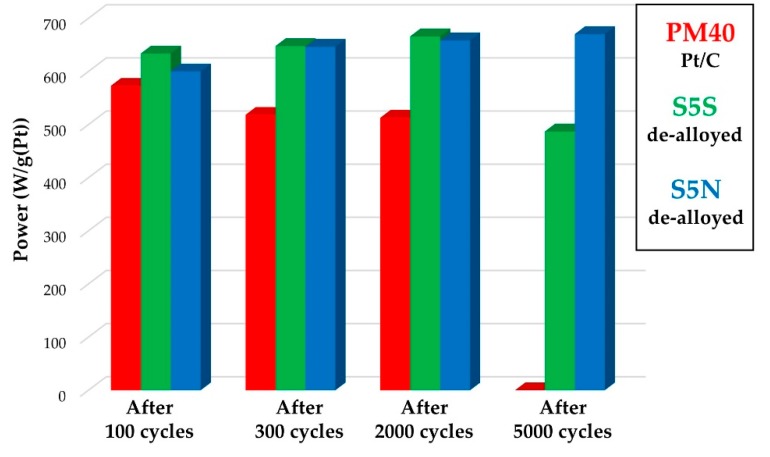
The maximum power of MEAs based on the studied PtCu/C and Pt/C catalysts after 100, 300, 2000 and 5000 cycles of stress testing in the membrane-electrode assembly.

**Table 1 nanomaterials-10-00742-t001:** Composition and structural characteristics of the obtained PtCu/C and commercial Pt/C materials.

Sample	Metal Component Composition of the Catalysts (According to XRFA)		Platinum Loading, ω (Pt), % wt.	2θ (111), grad	Average Crystallite Size, *D_Av_ **, nm
	In “as-Prepared” State	After Electrochemical Standardization			
S1	Pt_37_Cu_63_	Pt_67_Cu_33_	19.9 ± 0.5	41.2	3.0 ± 0.2
S2	Pt_31_Cu_69_	Pt_63_Cu_38_	23.4 ± 0.5	41.3	3.0 ± 0.2
S3	Pt_26_Cu_74_	Pt_67_Cu_33_	26.6 ± 0.5	41.2	2.8 ± 0.2
S1A	Pt_53_Cu_47_	Pt_63_Cu_38_	18.5 ± 0.5	40.9	2.7 ± 0.2
S2A	Pt_53_Cu_47_	Pt_67_Cu_33_	22.4 ± 0.5	41.0	3.0 ± 0.2
S3A	Pt_56_Cu_44_	Pt_67_Cu_33_	26.4 ± 0.5	40.8	2.7 ± 0.2
JM20 (Pt/C)	Pt	Pt	20.0 ± 0.5	39.9	2.0 ± 0.2

* *D_Av_*, nm—average crystallite size, calculated by the Scherrer formula.

**Table 2 nanomaterials-10-00742-t002:** ECSA values and parameters characterizing the catalysts behavior in the methanol electrooxidation reaction.

Sample	ECSA (*H_ads_*_/*des*_), m^2^/g(Pt)	ECSA (CO), m^2^/g(Pt)	*Q*_CH3OH_ * 10^2^, Cl/g(Pt)	*I*_max oxidation_ CH_3_OH, A/g(Pt)	Chronoamperometry Results at *E* = 0.70 V	
					*I*_initial_, A/g(Pt)	*I*_final_, A/g(Pt)
S1	39 ± 4	31 ± 3	127.8	970	492.3	230.6
S2	31 ± 3	33 ± 3	116.6	816	464.4	239.0
S3	37 ± 4	29 ± 3	70.1	600	432.3	82.8
S1A	38 ± 4	31 ± 3	96.0	868	529.5	248.1
S2A	43 ± 4	37 ± 4	107.8	997	578.7	269.9
S3A	33 ± 3	32 ± 3	72.3	647	411.7	117.7
JM20	81 ± 8	78 ± 8	32.9	350	291.6	126.5

* When calculating the specific values of the electric current and the amount of electricity, ECSA values were used, calculated by the adsorption/desorption of atomic hydrogen.

**Table 3 nanomaterials-10-00742-t003:** Composition and structural characteristics of alloyed PtCu/C catalyst (S4) and samples obtained after its treatment in various acids.

Sample	Composition of the Solution for Treatment	Composition of Metallic Component	Metals Loading, % wt.		Average size of Crystallites (XRD), nm
			Metals	Platinum	
S4	-	Pt_31_Cu_69_	23.4 ± 0.2	13.6 ± 0.2	2.8 ± 0.2
S4-1N	1M HNO_3_	Pt_57_Cu_43_	17.4 ± 0.2	13.9 ± 0.2	2.4 ± 0.2
S4-1Cl	1M HClO_4_	Pt_57_Cu_43_	17.2 ± 0.2	13.8 ± 0.2	2.6 ± 0.2
S4-1S	1M H_2_SO_4_	Pt_58_Cu_42_	17.8 ± 0.2	14.4 ± 0.2	2.5 ± 0.2
S4-5N	5M HNO_3_	Pt_63_Cu_37_	16.6 ± 0.2	13.9 ± 0.2	1.8 ± 0.2

**Table 4 nanomaterials-10-00742-t004:** Composition and parameters characterizing electrocatalysts electrochemical behavior.

Sample	Pt and Cu Ratio in the Sample after Completion of Electrochemical Measurements	ECSA, m^2^/g(Pt)	*E*_1/2_, V (1600 min^−1^)	Kinetic Current, *j_к_* (at *E* = 0.90 V)		*n*
				A/g (Pt)	A/m^2^ (Pt)	
S4	Pt_67_Cu_33_	39.4	0.90	161	4.1	3.8
S4-1N	Pt_65_Cu_35_	41.2	0.90	200	4.9	4.4
S4-1Cl	Pt_69_Cu_31_	35.7	0.89	130	3.6	4.0
S4-1S	Pt_68_Cu_32_	35.2	0.90	117	3.3	4.2
S4-5N	Pt_72_Cu_28_	39.2	0.90	140	3.6	3.5
JM20	Pt/C	81.8	0.89	156	1.9	4.0

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
