# Peer review of "Effective Platinum-Copper Catalysts for Methanol Oxidation and Oxygen Reduction in Proton-Exchange Membrane Fuel Cell"

_nanomaterials, 2020, doi:10.3390/nano10040742_

Round 1
Reviewer 1 Report
Dear Editor,
The manuscript “Effective Platinum-Copper Catalysts for Methanol Oxidation and Oxygen Reduction in PEM FC” reports on optimization of de-alloying procedure of Pt-Cu catalyst for improving their performances in MOR and ORR reactions, mainly limiting the detrimental copper dissolution during cell operation.
In my opinion, the topic investigated is of great interest and the results obtained can help to improve the fabrication of PEMFC especially from the durability point of view. The manuscript is however not well clearly written considering the data presentation. The absence of page and line numbers doesn’t help the reviewing process. Several sample labels are apparently different, like S1A and S1K or A1K, but, as far as I understand, they refer to the same material. The caption of figure 3 is not clear, probably there are typos in the (a, c), (c, d) or (b,d) assignments. Also, several typos are in the subscript/superscripts of chemical formulas and several sentences would need rewriting for clarity. The use of acronyms is, in my opinion, not appropriate. In the title I recommend to avoid “PEM FC” and write “proton exchange membrane fuel cell”. Moreover, in the introduction it is stated that PEMFC stands for hydrogen-air, that results somewhat confusing. De-alloyed materials should be better described in the introduction and some references should be given. In the materials and methods section several important details are missing, like solution concentrations, volumes, temperature and incubation times, etc.
Author Response
Thank you for carefully studying the article. The response to the comments is in the attached file.

Reviewer 2 Report
Comments:
The paper concerns the behavior of supported alloyed and de-alloyed platinum-copper catalysts, dispersed onto Vulcan XC72 carbon support for methanol oxidation and oxygen reduction in PEM FC. The subject is of importance from both a theoretical and practical point of view and appropriate for the journal. The authors have conducted a detailed study about the effect of the acid pretreatment of platinum-copper catalysts on their catalytic activity. Functional characteristics of platinum-copper catalysts were compared with those of commercial Pt/C catalysts when tested, both in an electrochemical cell and in H2/Air membrane-electrode assembly (MEA). The obtained results of the study are interesting and useful.
However, the authors should emphasize more clearly the new and original contribution of their research.
It is important to present TEM and XPS analysis, in terms of the change in morphology of the catalysts and the elemental state of the Pt and Cu on the surface before and after the acid pretreatment.
The results for the activity of the catalysts for MOR and ORR are not compared to those of the same kind of Pt(Cu)/C catalysts, obtained by faster and more efficient methods (for example by galvanic replacement).
Based on the above, I recommend publication after taking into account the comments.
Author Response

(The authors gave the same response as above.)

Reviewer 3 Report
This manuscript is well written. The reviewer recommends the publication of this manuscript.
Author Response
Thank you for carefully studying the article.
Reviewer 4 Report
This manuscript describes the de-alloyed PtCu/C electrocatalysts for MOR and ORR applications and its catalytic performance as cathodic catalysts in hydrogen/air PEMFCs. The authors ascribe the improvements to the de-alloying of “weakly bound” copper from the nanoparticles in acid treatment and electrochemical standardization. The experiments are carefully designed and details are thoroughly provided. There're some interesting points with the de-alloying using various acids and electrochemical cycling. The knowledge gained in this work could potentially provide reference for other researchers working in the field. I suggest the work is accepted for publication after addressing the following comments:
- A proof reading is required before submission. There’re many typos with the manuscript, e.g. the catalysts’ names should be kept coherent as S* not as A* in the discussion below Figure 4, 25oC rather than 23oC for CV analysis in experimental section, AFCBP bipotentiostat from Pine rather than Pain Research Instrumentation, ORR not ORP in the discussion of Table 4, the format of super and subscript, etc. The authors need to carefully correct all errors in the revision.
- The authors highlight the difference between the RDE and MEA techniques and the poisoning of Nafion through the manuscript, and more discussions could also be provided referring to ACS Energy Letters 2020, 4(9), 2104-2110, and the monitoring of mass activities of electrocatalysts could also be conducted in the MEA test for a comparison, for which the method details could be found in Applied Catalysis B: Environmental 2020, 260, 118031 & 2015, 164, 389-395.
- With Figure 4, a and c should be presented based on Pt mass, rather than normalization to ECSA which does not mean anything here. With b and d, all start points should be normalized to 100% to compare the stability of various electrocatalysts.
- In Figure 9, the curves should be named as polarization and power density curves rather than volt-ampere and watt-ampere characteristics. Why PM40 and S5 are used here, rather than PM20 and S4 as discussed through all sections above? Please put potential to the left axis and power to the right axis as commonly used in plotting these curves. The good power performance with cycling here could also be ascribed to the further de-alloying during the cycling process, not just the stability.
Author Response
A proof reading is required before submission. There’re many typos with the manuscript, e.g. the catalysts’ names should be kept coherent as S* not as A* in the discussion below Figure 4, 25oC rather than 23oC for CV analysis in experimental section, AFCBP bipotentiostat from Pine rather than Pain Research Instrumentation, ORR not ORP in the discussion of Table 4, the format of super and subscript, etc. The authors need to carefully correct all errors in the revision.
Thank you for carefully reading the article, typos corrected.
The authors highlight the difference between the RDE and MEA techniques and the poisoning of Nafion through the manuscript, and more discussions could also be provided referring to ACS Energy Letters 2020, 4(9), 2104-2110, and the monitoring of mass activities of electrocatalysts could also be conducted in the MEA test for a comparison, for which the method details could be found in Applied Catalysis B: Environmental 2020, 260, 118031 & 2015, 164, 389-395.
We are grateful for the recommendations regarding literary sources, but there are some inconsistencies in them. The first is the release of ACS Energy Letters, 2020, 4 (9). There is no such paper on the journal’s website yet, and an article with similar details published in 2019 in ACS Energy Letters 2019, 4, 9, 2104-2110 is called “Ionic Liquid-Modified Microporous ZnCoNC-Based Electrocatalysts for Polymer Electrolyte Fuel Cells” and is not related to the topic of our work. 2nd and 3rd recommended articles: Applied Catalysis B: Environmental 2020, 260, 118031: Peter Mardle, Xiaochao Ji, Jing Wu, Shaoliang Guan, Hanshan Dong, Shangfeng Du, “Thin film electrodes from Pt nanorods supported on aligned N-CNTs for proton exchange membrane fuel cells ”and Applied Catalysis B: Environmental, 2015, 164, 389-395, Yaxiang Lu, Shangfeng Du, Robert Steinberger-Wilckens, “Temperature-controlled growth of single-crystal Pt nanowire arrays for high performance catalyst electrodes in polymer electrolyte fuel cells." In these works, other materials were studied; the methodology for testing catalysts in the OIE was different. We do not consider it necessary to perform a second study using a different technique. The point is not in the specific methodology and in absolute values of the parameters, but in the differences which we have reliably established in the behavior of commercial Pt/C and prepared de-alloyed PtCu/C catalysts. In carrying out future research, we will take into account the methodological aspects of working with MEAs described in publications recommended for study.
With Figure 4, a and c should be presented based on Pt mass, rather than normalization to ECSA which does not mean anything here. With b and d, all start points should be normalized to 100% to compare the stability of various electrocatalysts.
We deliberately presented the results (Figure 4, a, c) as the dependence of the specific current on the potential. Table 2 shows the ECSA values for catalysts whose LSVs are shown in Figure 4 a, c. From these data it follows that ECSA Pt/C is about 2 times higher than ECSA PtCu/C catalysts. The transformation of Fig. 4a, c in the form of the dependence of the mass activity (A/g (Pt)) on the potential would actually be identical to a twofold increase in the current for Pt / C compared to the currents for PtCu/C. It is easy to see that the currents (A/g (Pt)) on Pt/C would still be significantly less than for PtCu / C catalysts. The ratio of the mass activities of the catalysts is also clear from the data in Table 2, where the current strength at the maximum of the LSV curve and the amount of electricity used to oxidize methanol are related to the mass of platinum, as the reviewer suggests. In Fig. 4 a, c, it was important for us to show that the specific activity per unit surface area of platinum is significantly higher in PtCu/C materials. This fact just demonstrates the positive effect of residual copper. This also means that when we or other researchers can get similar de-alloyed catalysts with a more developed surface (ECSA value close to Pt/C), their efficiency will increase even more. At the same time, taking into account the data of Table 2 related to the mass of platinum, as well as the simple equation: I (A/g (Pt)) = I (A/m2 (Pt)) * ECSA (m2/g (Pt)), any reader in Figures 4 a, c can easily assess the difference in activity related to the mass of platinum.
Regarding figures 4 b, d. The decrease in current is associated with poisoning of the surface of the catalysts by products of incomplete oxidation of methanol. The amount of oxidized methanol and, apparently, products of its incomplete oxidation per unit surface area of platinum, of course, is significantly larger on PtCu/C electrodes. Therefore, the current on these catalysts may decrease faster. If we imagine the current change in % of the initial value, this will create the illusion of poor quality of PtCu/C catalysts, their low tolerance. When we build the dependences for the absolute values of currents, we show that despite the high oxidation rates of methanol and the rapid accumulation of products, the activity of PtCu/C catalysts (A/m2) remains higher than for Pt/C. And this is very important, because when using catalysts in DMFC, it is high currents that are important, and not the degree of their deviation from the initial value. Moreover, it is easy to see from Figs. 4b, d that during the measurement time, the currents least decrease on the Pt/C catalyst. However, this is due precisely to the small amount of reacting methanol and, as a consequence, the low concentration of impurities that poison the electrode. I also note that for catalysts S1, S2, S1A and S2A, there is a tendency to slow down the decline and there is no reason to believe that the rate of methanol oxidation on them may become lower than on Pt/C.
In Figure 9, the curves should be named as polarization and power density curves rather than volt-ampere and watt-ampere characteristics.
We agree with the remark, made corrections.
Why PM40 and S5 are used here, rather than PM20 and S4 as discussed through all sections above?
S4 material and the de-alloyed catalysts obtained from it, containing 13-14% of the mass of platinum, were studied as ORR catalysts in an electrochemical cell. The platinum loading in the membrane-electrode block was supposed to be 0.4 mg/cm2. The use of catalysts containing 13% wt would lead to the formation of a thick catalytic layer and diffusion difficulties. Therefore, for the MEA test, it was decided to obtain analogues of the S4-N and S4-S catalysts with a higher loading of platinum close to 20% of the mass and compare their behavior with PM20 (20% wt Pt). As a result, S5 material was obtained, and from it de-alloyed catalysts S5N and S5S with a platinum loading of about 18.5% of the mass. Further, during the first test tests of the OIEs, it became clear that the PM20 material is rapidly degrading. Therefore, it was decided to compare the load tests of the MEAs based on S5N and S5S with the OIE based on the commercial Pt/C catalyst PM40 (40% wt Pt), which contains almost 2 times more platinum and is much more stable than PM20. In testing, we obtained an important result: de-alloyed PtCu/C catalysts containing 18% platinum only showed higher stability than PM40. Thus, we have shown that the MEA can be efficiently operated using a de-alloyed PtCu/C catalyst with a low platinum content but high stability. A similar result cannot be obtained using Pt/C catalysts even with 40% platinum loading, and even more so - with 20% Pt loading. Note that the tests of such catalysts in the MEA were carried out for the first time. However, the results obtained in this case correlate well with the data previously published by us, which showed that alloyed PtCu/C catalysts are characterized by very high stability during life time testing in the electrochemical cells [29, 49].
Please put potential to the left axis and power to the right axis as commonly used in plotting these curves.
We agree with the remark, made corrections.
The good power performance with cycling here could also be ascribed to the further de-alloying during the cycling process, not just the stability.
We agree with the reviewer that the selective dissolution of copper can continue during the cycling process. Speaking about the stability of the characteristics of membrane-electrode blocks containing de-alloyed PtCu/C catalysts, we do not conduct a detailed analysis of the reasons for such stability in this article. An analysis of the change in the composition of PtCu/C catalysts during the cycling process was performed by us earlier in [49], the results of which are referred in this article. Moreover, it was shown in [49] that the change in the copper content in the catalysts after stress testing is small. In the context of this new article, it is important that the stable operation of de-alloyed PtCu/C catalysts in the MEA does not lead to poisoning of Nafion and a decrease in the functional characteristics of the MEA.

Round 2
Reviewer 2 Report
The authors did not take into account all the recommendations made. The answer is incomplete and therefore I cannot recommend publication in the present state.
Author Response
We apologize for the inattentive reading of the reviewer comment for the first time (we did not saw the comments that “jumped” to another page). This was an unfortunate accident. Sorry!
It is important to present TEM and XPS analysis, in terms of the change in morphology of the catalysts and the elemental state of the Pt and Cu on the surface before and after the acid pretreatment.
In this article, we refer to our previously published results, according to which PtCu / C catalysts in the “as received” state contain part of copper in the form of oxides, and part as a metal component of PtCu nanoparticles. In particular, on page 5, lines 18-19, we write: “Apparently, a larger or a smaller fraction of copper in these samples is contained in X-ray amorphous oxides [45].” In lines 33-35 on the same page, we indicate that “A decrease in the concentration of copper in the samples cannot be associated only with the dissolution of its amorphized oxides, since judging by the data of the X-ray diffractometry, the content of metallic copper in the solid solution based on platinum, decreases. This leads to the observed decrease in the 2-theta angle for the maximum (111) during the transition from the initial to the acid-treated samples (Table 1, Figure 1).” In principle, such differences in the composition of PtCu nanoparticles before and after treatment are well known (studied). Moreover, for almost all bimetallic systems (d-metal - Ni, Co, Fe, Ag, Cu, etc.) it is known that after treatment in acids or electrochemical standardization, the nanoparticles acquire a secondary core-shell structure, in which their surface is enriched with platinum , and the atoms of the alloying component are mainly localized in the inner parts of the nanoparticles.
From the point of view of the general analysis of the composition, XPS will show approximately the same as that shown by X-ray fluorescence spectroscopy, which we applied in the work. He will also show that in materials in the “as prepared” state a significant part of copper is contained in oxides. But we already know this and refer to previously published results [45].
Nevertheless, if a respected reviewer considers such a study to be fundamentally important for the article, then we will conduct it. Unfortunately, this may take about 20 days, since the university’s laboratories are now quarantined due to coronovirus. (We hope that quarantine will be lifted from April 10).
A TEM study of such alloyed PtCu/C catalysts was carried out by us in [22], the results of which are repeatedly referred to in the text of this article. The use of TEM to study changes in the morphology of catalysts as a result of acid treatment will not give the desired result, since the visual changes will not be significant. An example is the work [50], where the authors were able to see small changes in the composition and structure of PtCu nanoparticles as a result of processing. But i) in [50], very large PtCu nanoparticles were studied (in the catalysts synthesized by us, they are much smaller, see [22]); ii) the authors of [50] studied the same nanoparticles, that is, they conducted an in situ study. We do not have such an opportunity.
The results for the activity of the catalysts for MOR and ORR are not compared to those of the same kind of Pt(Cu)/C catalysts, obtained by faster and more efficient methods (for example by galvanic replacement).
A direct comparison of the results (absolute parameter values) characterizing the activity of the catalysts for MOR and ORR is not entirely correct. Different authors study the catalytic layers of different thicknesses, add different amounts of Nafion to them. The composition and amount of catalytic ink applied to the electrode also varies. Therefore, in the literature you can find a variety of activity values. We took a different, more correct path. We compare PtCu/C catalysts with well-known commercial Pt/C catalysts, which generally show good performance. We prepare all catalytic layers in the same way and can reliably state how many times the activity of our PtCu/C catalysts is higher than that of the reference catalysts. Many authors also come using commercial Pt/C catalysts as unique reference samples.
